# Towards Universal Neural Operators through Multiphysics Pretraining

## Abstract

Although neural operators found common use in contemporary data-driven physical systems simulation, their training procedure remains computationally expensive and time-consuming. Some advances have been made with the study of downstream problems, where the model is trained on a simpler problem and later fine-tuned on a more challenging one to achieve better quality and lower over-all time costs. In this research we examine capabilities of transformer-based neural operator architectures, which were previously used only for particular problems solutions, in more generalized transfer learning. We evaluate performance of the transformer- and state space model-based neural operators on wide range of downstream PDE simulation problems, including extension of models to the out-of pretraining sample parameter values, addition of new variables into the dynamics, and transfer of operator, trained on datasets, composed of solutions of several differential equations. The results indicate the ability of neural operators of advanced architectures to be used to transfer knowledge between problems, involving partial differential equations.

## 1 Introduction

Contemporary science commonly uses partial differential equations (PDEs) and systems of partial differential equations to model spatio-temporal processes. For instance, reaction-diffusion equations describe, how mass moves and disperses within a fluid system, and gas and liquid dynamics are commonly modeled with variants of Euler/Navier-Stokes equations. While the analytical solutions are applicable in idealized problem statement, it is challenging to construct them in realistic scenarios. Thus, numerical simulation techniques, such as finite-element or spectral methods have been developed, yet in many cases such solutions can be computationally costly. However, in many cases, such as meteorological forecasting or multi-physics simulation in the engineering, the numerical solution of differential equations tends to be a computationally costly procedure.

With the development of scientific machine learning, greater emphasis in physics systems simulations is placed on data-driven methods. Physics-informed neural networks (PINN) (Raissi et al., 2019) extend the loss with PDE-based terms, so the training has an objective of matching network's output with governing PDE. However, PINNs require explicit formulations and guarantee accuracy only at training mesh nodes. Operator learning, realized through Deep Operator Networks (DeepONet) (Lu et al., 2021) and kernel-based neural operators (NO) (Kovachki et al., 2023), offers a faster alternative to classical solvers, approximating mappings between functional spaces rather than dynamics at discrete nodes, providing discretization invariance and efficient inference.

A recent direction in scientific machine learning is the design of foundation models. Originating in natural language processing (Bommasani, 2021) and vision-language models (Awais et al., 2025), such models contain billions of parameters (e.g., GPT-4 exceeds one trillion (Achiam et al., 2023)) and get pretrained on large-scale datasets. They can then be fine-tuned for downstream tasks at reduced cost. In this work, we aim to develop a standardized approach to applying large neural operator - based models to generalize dynamics and transfer learning across varying PDE problems. Thus, with the neural operator the model is pre-trained on a simplified problem statement, which is later transferred to another (typically, more complex) problem in the fine-tuning phase.

Another key element in the development of pretraining approaches and solution of subsequent downstream problems in scientific machine learning is popularization of transformer-based neural

operator architectures and introduction of self-supervised learning paradigms. However, the highest quality of pretraining and fine-tuning with existing architectures is still associated with cases, when the downstream problems are similar to the original pretraining ones (Zhou et al., 2024). In this study we propose a neural operator - based model with improved generalizing ability, that can be developed into a foundational model for scientific ML problems. Furthermore, to decrease fine-tuning costs, we introduced adapters with reduced parameters count.

**Our contributions are as follows:** the proposed method enables neural operator learning on diverse multi-physics datasets by introducing an adapter-based approach for simultaneous training on PDE-based problems with different sets of input functions. The results demonstrate that the transfer learning approach can significantly enhance model quality and reduce fine-tuning costs.

**Code and data** are available in the GitHub repository, anonymized for review purposes: `https://anonymous.4open.science/r/multiphysics_neuroperators_pretraining-9467/`.

## 2 RELATED WORK

Pretraining of neural operators has thus far been primarily case-specific, with limited generalization and without knowledge transfer between multi-physics scenarios. Initially, the concept of transfer learning using DeepONet operator models are employed in conditional shift scenarios in study (Goswami et al., 2022). Also, issues of solving transfer learning problems on multi-scale data with convolutional neural networks were discussed in (Subel et al., 2023). Foundational models in form of General Physics Transformers, capable of accurately predicting governing dynamics without significant finetuning, were proposed in the study (Wiesner et al., 2025).

Several foundational models for PDE systems have been proposed beyond classical NO approaches: a foundational model tranining framework for equations of fixed types (in the study, steady-state equations were considered) is proposed in the research (Subramanian et al., 2023). Boundary-Embedded Neural Operators (BENO) (Wang et al., 2024) solve elliptic PDEs using graph neural networks, where boundary geometry is encoded via a transformer block into latent vectors guiding message passing. Other works leverage transformer-based architectures to encode PDE structure (Zhang et al., 2024).

Transformer-based approaches proved to be capable of modeling complex interactions between selected token functions. POSEIDON, a hierarchical vision transformer with shifted windows, applicable to transfer knowledge across Euler/Navier–Stokes cases (Herde et al., 2024). Codomain Attention Neural Operator (CoDA-NO) (Rahman et al., 2024), designed for multiphysics PDE transfer learning, employs codomain attention with function space dot product.

The topic of transformer-based architectures within neural operator learning has been explored in study (Boya & Subramani, 2024). Despite some promising results, presented in , we deliberately avoid physics-informed approaches and focus on assessing the capacity of neural operators to learn generalized dynamics purely from data samples.

## 3 METHOD

Operator learning framework has been developed for data-driven modeling of dynamical systems, governed by parametric partial differential equations $L_p u(t, \mathbf{x}) = f(t, \mathbf{x})$ on the bounded domain $\Omega$ in a mesh-agnostic (albeit with some limitations, as examined in (Fanaskov & Oseledets, 2023)) approach. While the operator learning can be extended to the inverse problems, where the objective is to discover the coefficients of the known equation from the data, we limit our scope to the direct problems - approximation of solution, that is the discovery of mappings between the Hilbert spaces of input and output functions.

**Neural operator learning:** In this research, we have focused on the kernel-integral neural operators, mainly Fourier Neural Operators. Neural Operators are designed to learn mappings between the input space $\mathcal{U}$, representing sets of input functions $\mathbf{a} = \{a_1, \ldots a_{n\_in}\}$, $a_i : \mathcal{D} \longrightarrow \mathbb{R}$ and the output functions $\mathbf{u} = \{u_1, \ldots u_{n\_out}\}$, $u_i : \mathcal{D} \longrightarrow \mathbb{R}$. While the outputs are typically fixed to the

dependent variables of the PDE (or system of PDEs), the choice of input functions is guided by the equation structure and includes meshes, initial conditions, forcing terms, or equation coefficients.

To respect the non-localities of the model, NO design adds integral kernel operators $(\mathcal{K}(v))(x)$ to the arguments ($A_t$, and $b_t$ for weights and biases) of activation functions $\sigma$. Here the $\kappa_t \in C(D_{t+1} \times D_t | \theta_{k,t})$; $\theta_{k,t} \in \mathbb{R}^{n_{v_{t+1}} \times n_{v_t}}$ is the kernel function, parameterized (with $\theta_{k,t}$) by the method of choice (FNO, GNO, etc.), and $v_t(y)$ - hidden representation of input functions, obtained from the $t$-th layer. $D_t$ is the hidden dimensionality, which is linked to the number of modes in FNO. Thus, the parameters of NO main part include weights, biases and kernel parameters $\theta_{\mathcal{F}} = \{A_t, b_t, \theta_{k,t} : t = 1, \, ... \, , n_{\text{layers}}\}$.

$$\mathcal{F}_t(x) = \sigma \left( A_t v_t(x) + \int_{D_i} \kappa_t(x,y) v_t(y) dy + b_t(x) \right), \ \forall x \in D_t, \ t = 1, \, ... \, , n_{\text{layers}}. \quad (1)$$

The architecture of layers sequence in NO goes as follows: the inputs are transformed to their higher-dimensional hidden representation by lifting layers (typically, a feed-forward neural network) $\mathcal{L}$ : $\mathcal{L}(\mathbf{a}) = \sigma \left( A_{\mathcal{L}} \mathbf{a} + b_{\mathcal{L}} \right)$, where parameters of the lifting include weights and biases $\theta_{\mathcal{L}} = \{A_{\mathcal{L}}, b_{\mathcal{L}}\}$. The hidden representations are sequentially mapped with the integral-operator blocks equation 1. In contrast, the output of the last block is projected to the space of outputs by the point-wise function (represented by a feed-forward neural network) $\mathcal{P}$ with parameters $\theta_{\mathcal{L}} = \{A_{\mathcal{P}}, b_{\mathcal{P}}\}$. The operator approximation takes form of model $\mathcal{G}_\theta$:

$$\tilde{\mathbf{u}}(\mathbf{x}) = \mathcal{G}_\theta(\mathbf{a}) = \mathcal{P} \, \circ \, \mathcal{F} \, \circ \mathcal{L}(\mathbf{a}) = \, \mathcal{P} \, \circ \, \mathcal{F}_{n\_layers} \, \circ \, ... \, \circ \, \mathcal{F}_1 \, \circ \, \mathcal{L}(\mathbf{a}). \quad (2)$$

Having the neural operator model, the training procedure involves minimization of selected loss functional $L$ between the ground truth in the points of sample $\mathcal{S}$ and the operator prediction with the respective inputs $\mathcal{G}_\theta(\mathbf{a})$. For example, commonly used MSE (Mean Squared Errors) metric will be written as:

$$L_{MSE}(\theta; \mathcal{S}) = \frac{1}{|\mathcal{S}|} \sum_{(\mathbf{a},u) \in \mathcal{S}} \frac{1}{|\mathcal{D}|} \sum_{(x_j,t_k) \in \mathcal{D}} \left( \mathcal{G}_\theta(\mathbf{a}(x_j, t_k)) - u(x_j, t_k) \right)^2. \quad (3)$$

**Improving the generalization ability of neural operators:** In this research, we employed two types of NO modifications: state-space models and transformer-based models. In the first approach, inserting a Mamba-SSM module (Gu & Dao, 2023) $\mathcal{M}_\phi$ after the lifting map $\mathcal{L}$ allows the model to encode long-range temporal and spatial dependencies directly in the hidden representation. For lifted features $v_0(x) = \mathcal{L}(\mathbf{a})(x)$, the Mamba module computes

$$\widetilde{v}_0(x,t) = (\mathcal{M}_\phi v_0)(x,t) = \sum_{\tau \leq t} K_\tau \, v_0(x, t - \tau), \quad (4)$$

with learnable convolution kernels $K_\tau$ defining the causal recurrence. This step acts as a latent preconditioner: embeddings are aligned with dominant dynamical motifs (transport, diffusion, oscillation) common across PDEs, so that when passed into the Fourier integral layers, the effective operator acts on inputs of reduced variability and lower spectral rank. Consequently, the composition $\mathcal{F}_t \circ \mathcal{M}_\phi$ yields more stable training and improves efficiency in transferring pre-trained representations to new PDEs during fine-tuning.

The next approach, examined in the study, involved the attention method & transformer-based blocks. The introduction of Perceiver (Jaegle et al., 2021b) enabled the encoding of information with a smaller number of latent feature arrays, internal to operator blocks, thereby operating with more abstract feature arrays and maintaining a limited number of parameters. As the operators we employ, we use blocks based on the Perceiver IO (Jaegle et al., 2021a), where the mapping is performed with a symmetrical cross-attention mechanism for outputs, which mirrors the cross-attention block for constructing representations of latent arrays from the input process. While the previously used self-attention blocks can discover dependencies between hidden features, obtained from lifting or previous layers, Perceivers are able to constructs additional latent process representation.

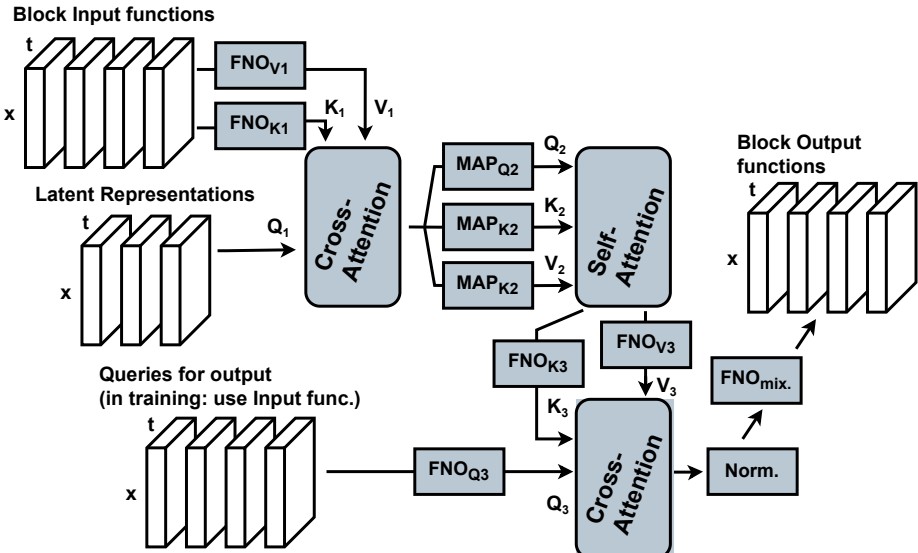

Figure 1: Scheme of the Perceiver-IO block with neural operators, employed to produce keys, queries and values for the cross attention. Here, MAP denotes feed-forward neural networks or Fourier neural operators.

The latent variables and input state are taken as tokens for the first with the cross-attention block, where keys and values are obtained from FNO-based mapping from the inputs $K_1 = FNO_{K_1}(X)$, $V_1 = FNO_{V_1}(X)$, and latent variables are taken as queries $Q_1 = L$. The cross-attention block is followed by self-attention between latent representation. The output of the block is constructed with the cross-attention, matching the queries from the inputs with the keys and values, taken from the transformed latent representations.

Commonly used self-attention mechanism involves similarity function $\mathrm{sim}(q_m, k_j)$ between given sets of finite-dimensional vectors of queries $\{\mathbf{q}_i\}$, $i = 0, \ldots, N_q$ and keys $\{\mathbf{k}_i\}$, $i = 0, \ldots, N_k$, which is used to obtain the output from the $m$-th query with the set of value vectors $\{\mathbf{v}_i\}$, $i = 0, \ldots, N_v$ with the relation, presented in Eq. 5.

$$\mathrm{Attn}_m(Q, K, V) = \frac{\sum_j \mathrm{sim}(\mathbf{q}_m, \mathbf{k}_j)\mathbf{v}_j}{\sum_j \mathrm{sim}(\mathbf{q}_m, \mathbf{k}_j)},$$
$$\mathrm{sim}(\mathbf{q}_m, \mathbf{k}_j) = \exp\left(\frac{\mathbf{q}_m^T, \mathbf{k}_j}{\tau}\right). \tag{5}$$

In the FNO-based attention mechanisms queries, keys and values $Q, K, V$ are obtained by applying several stacked neural operators layers (blocks) with trainable sets of parameters. The similarity is commonly obtained using *Softmax* function of the dot products between corresponding queries and keys. Codomain attention mechanisms, introduced in (Rahman et al., 2024), have and advantage to the conventional attention blocks in the neural-operator based problems: the dot product detecting similarity not between samples, but between features, mapped with neural operators. The scheme of the implemented Perceiver-based operator is presented on the Fig. 1.

**Pre-training and fine-tuning:** One of the benefits of using lifting-operator-projection architecture is the simplicity of decoupling adapters from the main model, streamlining model storage and extension for novel fine-tuning problems. The `lift` and `proj` blocks are considered as the adapters, representing the mappings, associated with the problem-specific part of dynamics: they are introduced to contain different cardinality input sets, projecting into the fixed number of hidden features and contain small number of parameters to represent limited part of the total model variance, as it is common in the adapter design for large language models (Hu et al., 2022).

In the pre-training phase the entire parameters set $(\theta_{\mathcal{P}_1}, \ldots, \theta_{\mathcal{P}_N}, \theta_{\mathcal{F}}, \theta_{\mathcal{L}_1}, \ldots, \theta_{\mathcal{L}_N})$ is subject to optimization. By problems 1 to $N$, we present separate physical processes, demanding different (but, probably, overlapping) sets of input functions. Previously, such problems were solved with liftings with extensive inputs. For example, in training on the set of steady-state problems, coefficients before terms with all spatial derivatives, which occur in the training set, are used as inputs. In the fine-tuning stage we fix the parameters $\theta_{\mathcal{F}}$ both to highlight the generalizing properties of the operator and to reduce training costs: only the new adapter parameters $(\theta_{\mathcal{P}_{ft}}, \theta_{\mathcal{L}_{ft}})$ are trained.

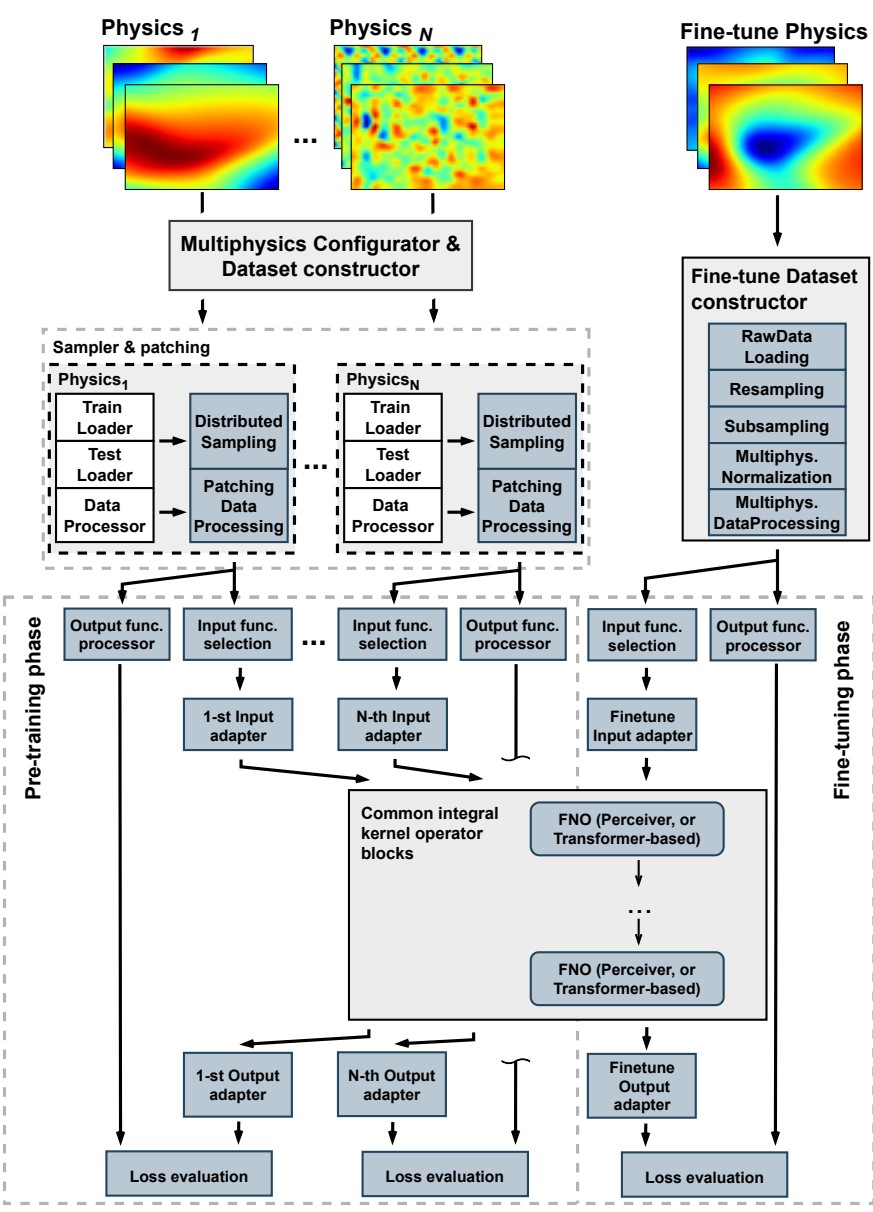

Figure 2: Scheme of the neural operator pre-training and fine-tuning stages. In this scheme, FNO blocks denote arbitrary kernel integral operators, including the Perceiver-, transformer-based neural operators or similar architectures. The separate physics 1 to $N$ may vary from the different manifestations of the same system to multiple different physics (but with the same problem dimensionality) in the dataset collection.

## 4 EXPERIMENTS

We aim on validating our approach on three distinct types of problems without modifications in the modeling approach. The first scenario involves cases, when the pre-training and fine-tuning processes are governed by the equations with same input functions, but with different parameters. In addition to parametric variations within a single physical law, we further extend our investigation to cross-domain transfer learning between disparate classes of differential equations representing distinct physical phenomena. The scenario assumes that the pre-training and fine-tuning processes are driven by fundamentally different physical laws, each of which, in turn, may have is's own parametric variations. In this context, we investigate the ability of the model to identify and exploit deep mathematical invariants and structural regularities common to different classes of differential equations.

To demonstrate the effectiveness of our method, we selected several datasets representing diverse physical phenomena including advective transport processes, nonlinear wave dynamics governed by Burgers equation, reaction-diffusion systems exhibiting pattern formation, and other fundamental physical processes described by partial differential equations. A detailed description is provided in Appendix A.2.

$$\text{NMAE}(\theta) = \frac{1}{|\mathcal{D}_{\text{RD}}^{\text{test}}|} \sum_{(\mathbf{a}, u) \in \mathcal{D}_{\text{RD}}^{\text{test}}} \frac{\|\mathcal{G}_\theta(\mathbf{a}) - u\|_{1,G}}{\max_G u - \min_G u + \varepsilon}. \tag{6}$$

In this comparison, we employ developed post-lifting MambaFNO models, post-lifting LocalAttnFNO models, and Perceiver IO-based neural operators. As the baseline, we employed the default FNO. We used the range-normalized mean absolute error (NMAE) on grid $G$ equation 6 as a quality metric.

As the references for comparing the algorithm performance, we have employed default Fourier neural operators as proposed in (Kovachki et al., 2023), codomain attention neural operators from (Rahman et al., 2024) in both pretraining and fine-tuning modes, and Swin v2 in the manner, proposed by (Herde et al., 2024).

**Out-of-sample parameter values scenario** First, we conducted several experiments on cases where the pretraining equations and fine-tuning ones differed only in the coefficient values. The experiments were conducted using Burgers' equation (where the kinematic viscosity was varied between pretraining and fine-tuning), the Gray-Scott model of the reaction-diffusion process (the diffusion rates and sinks were altered), and the Navier-Stokes equations for an incompressible flow. In this scenario, the adapters are transitioned from pre-training to fine-tuning parameters, using the previously optimized parameter values as initial guesses. The results of the comparison are presented in Tab. 1.

Table 1: Average metric values for out-of-sample parameter values across all conducted experiments. Methods were compared with training from scratch on the examined datasets scenario and using the pre-trained model to fine-tune on the new dynamics.

| Model | MSE | NMAE (%) | Avg. epoch (s) | Param. |
|---|---|---|---|---|
| Mamba FNO (pretr.) | $1.009 \times 10^{-7}$ | 0.0120 | 21.91 | $\approx 10^7$ |
| Mamba FNO (scratch) | $1.193 \times 10^{-7}$ | 0.0213 | 40.14 | $\approx 10^7$ |
| Perc. (pretr.) | $1.425 \times 10^{-7}$ | 0.0169 | **3.21** | $\approx 10^8$ |
| Perc. (scratch) | $1.981 \times 10^{-7}$ | 0.0219 | 204.73 | $\approx 10^8$ |
| FNO (scratch) | $1.774 \times 10^{-7}$ | 0.0204 | 7.44 | $\approx 10^6$ |
| Swin-v2 (p.+s.) | $4.391 \times 10^{-8}$ | **0.0092** | 101.3 | $\approx 10^9$ |
| CoDA-NO (pretr.) | $2.881 \times 10^{-7}$ | 0.0343 | 62.91 | $\approx 10^8$ |
| CoDA-NO (scratch) | $4.912 \times 10^{-7}$ | 0.0712 | 63.29 | $\approx 10^8$ |

**Input function set extension scenario** To assess the applicability of the adapter-based approach to more complex problems, several experiments were conducted on scenarios where the equations

in fine-tuning datasets were extended with additional terms. Such extension can be illustrated by Gray-Scott model of the reaction-diffusion process, involving two reacting chemical species 10. Their concentrations are denoted as $u$ and $v$; corresponding diffusion rates are $r_u$ and $r_v$; while $b$ and $c$ represent substances sinks. Nabla symbol $\nabla$ denotes gradient, while $\nabla^2$ symbolic operator denotes sum of second order represents partial derivatives along spatial axes in the domain.

$$\begin{cases} \frac{\partial u}{\partial t} = r_u \nabla^2 u - uv^2 + b(1+u), \\ \frac{\partial v}{\partial t} = r_v \nabla^2 v - uv^2 + (b+c)v. \end{cases} \quad (7)$$

As in other validation cases, the equation was modified by adding the convective component of the dynamics: in 2-dimensional Gray-Scott model $x$ and $y$ components of the velocity $w$ are represented by $w_x$ and $w_y$:

$$\begin{cases} \frac{\partial u}{\partial t} + w \cdot \nabla u = r_u \nabla^2 u - uv^2 + b(1+u), \\ \frac{\partial v}{\partial t} + w \cdot \nabla v = r_v \nabla^2 v - uv^2 + (b+c)v. \end{cases} \quad (8)$$

The aggregated results of the experiments are presented in Tab. 2.

Table 2: Average metric values for out-of-sample parameter values across all conducted experiments. Methods were compared with training from scratch on the examined datasets scenario and using the pre-trained model to fine-tune on the new dynamics.

| Model | MSE | NMAE (%) | Avg. epoch (s) |
|---|---|---|---|
| Mamba FNO (pretr.) | $3.91 \times 10^{-6}$ | **0.0041** | 131.2 |
| Mamba FNO (scratch) | $4.291 \times 10^{-6}$ | 0.0054 | 261.1 |
| Perc. (pretr.) | $4.107 \times 10^{-6}$ | 0.0051 | **20.4** |
| Perc. (scratch) | $6.315 \times 10^{-6}$ | 0.0074 | 804.0 |
| FNO (scratch) | $7.286 \times 10^{-6}$ | 0.0121 | 41.3 |
| Swin-v2 (p.+s.) | $6.276 \times 10^{-6}$ | 0.009 | 301.1 |
| CoDA-NO (pretr.) | $1.043 \times 10^{-5}$ | 0.013 | 185.1 |
| CoDA-NO (scratch) | $1.239 \times 10^{-5}$ | 0.018 | 181.9 |

**General multi-physics learning** In the final stage, we evaluated the capabilities of the developed methods to transfer knowledge from the dynamics of advection and Burgers' equation to reaction–diffusion, based on the corresponding datasets from the PDEBench dataset (Takamoto et al., 2022). The combined results of the experiments are presented in Tab. 3. The significant speedup achieved with the pre-trained models can be attributed to the optimization of just a subset of parameters, whereas "from scratch" involved a full parameter search.

To enhance performance of the model training in the multi-physics scenario due to the prohibiting costs of using expanded datasets we have employed techniques of gradient checkpointing (Chen et al., 2016) and variation of Adam optimizer, based on the Gradient Low-Rank Projection method (GaLore) Zhao et al. (2024), while examples of training are presented on Fig. 3.

## 5   CONCLUSION

In this research, we have examined the performance neural operators architectures, based on transformers and structured state space models, on a set of transfer learning problems, involving change of modeled equations' parameters, inclusion of additional physics, and, finally, pretraining on multiphysics datasets. In contrast to the default NO approach, studied models have greater generalizing power: extensive usage of latent parameter sets and problem-specific adapters allows easier transfer of knowledge to significantly reduce cost of obtaining model of decent quality even for novel PDE-based problems.

Table 3: Results of experiments with Heat & Reaction–Diffusion equation extension, and multi-physics pre-training with fine-tuning on different dynamics.

| Model | MSE | NMAE (%) | Avg. epoch time (s) |
|---|---|---|---|
| PL MambaFNO (pretr.) | $1.415 \times 10^{-7}$ | **0.0174** | 644.0 |
| PL MambaFNO (scratch) | $2.718 \times 10^{-7}$ | 0.0258 | 956.8 |
| PL LocalAttnFNO (pretr.) | $2.313 \times 10^{-7}$ | 0.0245 | 2413.0 |
| PL LocalAttnFNO (scratch) | $3.129 \times 10^{-7}$ | 0.0298 | 2285.0 |
| NO Perceiver (pretr.) | $1.425 \times 10^{-7}$ | 0.0189 | **98.0** |
| NO Perceiver (scratch) | $1.981 \times 10^{-7}$ | 0.0298 | 4618.0 |
| FNO (scratch) | $2.508 \times 10^{-7}$ | 0.0240 | 149.0 |
| Swin-v2 (p.+s.) | $8.391 \times 10^{-7}$ | 0.142 | 2071.0 |
| CoDA-NO (pretr.) | $2.913 \times 10^{-7}$ | 0.0403 | 2108.4 |
| CoDA-NO (scratch) | $3.041 \times 10^{-7}$ | 0.0441 | 2144.4 |

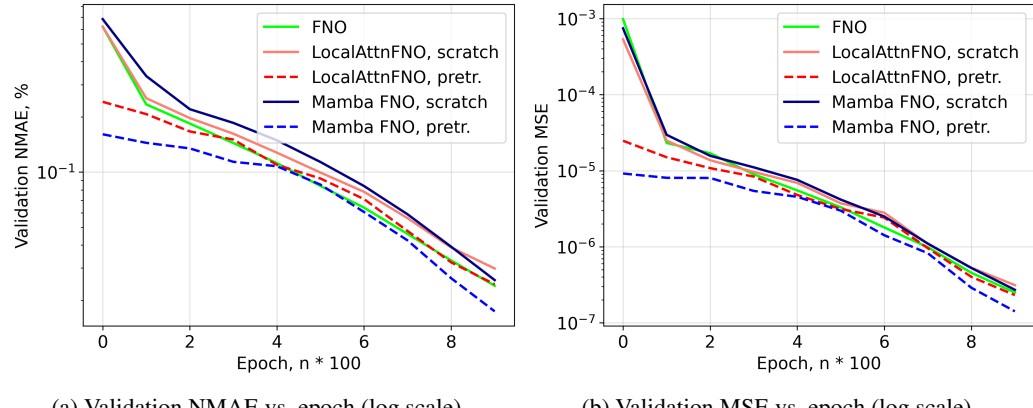

(a) Validation NMAE vs. epoch (log scale).  (b) Validation MSE vs. epoch (log scale).

Figure 3: Training dynamics on Reaction–Diffusion fine-tuning: comparison of architectures under pretraining and no-pretraining.

This study was conducted primarily as the first stage of work towards a foundational model, pre-trained on vast and heterogeneous multiphysics dataset collections. Additionally, the methodology of detecting the sets of optimal equations, that are best combined together into the pre-training datasets, shall be developed to further improve model quality. The follow-up work shall be directed towards improving model technical performance and training, implementing data augmentation tools (both generic and PDE-specific, e.g., based on Lie symmetries), and further examining neural operator generalizations.

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

## A DATA DESCRIPTION

### A.1 DATA PREPROCESSING FOR MULTIPHYSICS

To enable training neural operators on a wide spectrum of physical problems, we introduce a unified preprocessing pipeline (see Fig. 2). The core idea is to abstract away differences between individual datasets and physical laws, providing the model with a standardized representation of the problem.

Unlike conventional preprocessing pipelines designed for a single PDE family, our approach is explicitly multiphysics-oriented. This means that we can seamlessly combine:

- Intra-equation variability, e.g. multiple parameterizations of the same PDE (different permeability coefficients in Darcy flow, viscosity values in Burgers' equation, or boundary conditions in elasticity).
- Inter-equation diversity, where datasets originate from fundamentally different physical laws (e.g., elliptic PDEs such as Darcy flow, parabolic PDEs like heat conduction, and hyperbolic systems such as Euler or Navier–Stokes).

The preprocessing consists of several modular components:

- RawDataLoader, which imports raw simulation data in heterogeneous formats (e.g. .pt, .h5).
- Resampler, which projects all input-output functions from their native grids (differing in type and resolution) onto a shared, predefined grid of constant resolution via interpolation. This ensures the neural operator receives inputs of identical dimensionality, independent of the original PDE's discretization.
- Subsampling, which adapts data resolution to the operator's requirements while reducing redundancy.
- MultiphysNormalizer, which stores dataset-specific statistics and ensures normalized feature ranges for stable training.
- MultiphysDataProcessor, which encapsulates task-dependent routines such as encoding inputs/outputs, enforcing boundary masks, or applying domain-specific transformations.

These modules are combined within the MultiphysicsDataset and orchestrated through a task-specific configuration dictionary. This design allows researchers to define new PDEs or parameter regimes simply by extending the configuration file, without modifying the training loop. The pipeline thus provides a scalable mechanism to unify disparate physical domains and enables effective training of multiphysics neural operators.

### A.2 DATASETS

To evaluate the proposed framework, we conduct experiments on several representative datasets spanning different physical domains. These datasets cover fluids, porous media, and nonlinear wave phenomena, thereby providing a challenging multiphysics benchmark. Below we briefly describe each dataset.

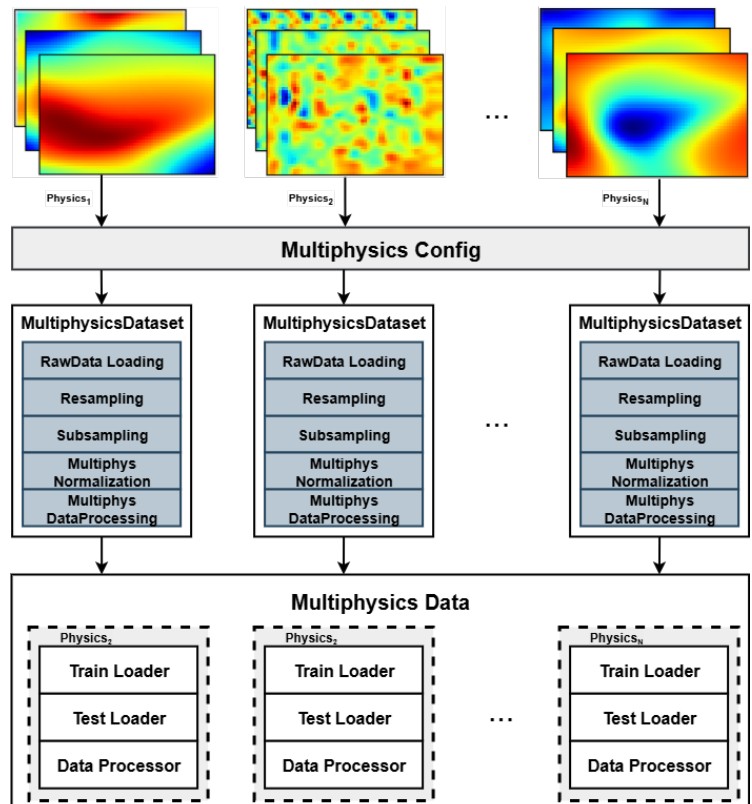

Figure 4: Multiphysics Configurator and Dataset constructor

*Advective*: A hyperbolic PDE modeling the transport of a quantity (e.g., heat or pollutant) by a velocity field. The input is the initial concentration field and the velocity profile, while the output is the concentration distribution over time. This benchmark tests the operator's ability to handle wave-like propagation and advection-dominated phenomena.

$$\frac{\partial u}{\partial t} + \mathbf{v} \cdot \nabla u = 0 \tag{9}$$

*Diffusion-Reactions*: A parabolic PDE, modeling processes such as chemical reactions and population dynamics.

$$\begin{cases} \frac{\partial u}{\partial t} = r_u \nabla^2 u - uv^2 + b(1+u), \\ \frac{\partial v}{\partial t} = r_v \nabla^2 v - uv^2 + (b+c)v. \end{cases} \tag{10}$$

The input is the initial concentration and the reaction rate parameters, while the output is the concentration evolution over time. This benchmark evaluates the operator's capacity to capture coupled diffusion and nonlinear reaction effects.

*Darcy2D*: A second-order elliptic PDE defined on a unit square.

$$-\nabla \cdot (a(x)\nabla u(x)) = f(x) \tag{11}$$

The input function is the spatially varying permeability field, and the task is to predict the pressure solution u. This dataset tests the operator's ability to handle highly heterogeneous coefficients.

*Burgers'*: A one-dimensional nonlinear PDE capturing shock formation and dissipation effects.

$$\frac{\partial u}{\partial t} + u\frac{\partial u}{\partial x} = \nu\frac{\partial^2 u}{\partial x^2} \tag{12}$$

The input is the initial velocity profile, and the model predicts its temporal evolution. This dataset emphasizes nonlinear dynamics and stability.

*Navier–Stokes*: A time-dependent system describing incompressible viscous fluids.

$$\begin{cases} \frac{\partial \mathbf{u}}{\partial t} + (\mathbf{u} \cdot \nabla)\mathbf{u} = -\nabla p + \nu\Delta\mathbf{u} + \mathbf{f} \\ \nabla \cdot \mathbf{u} = 0 \end{cases} \tag{13}$$

The input consists of initial vorticity frames, while the task is to predict subsequent frames of the velocity field. This benchmark evaluates temporal prediction and turbulence modeling.

*Heat Conduction*: A parabolic PDE modeling temperature diffusion across heterogeneous media.

$$\frac{\partial u}{\partial t} = \nabla \cdot (k(x)\nabla u) \tag{14}$$

The input is the initial heat distribution and conductivity, while the output is the temperature profile over time. This benchmark highlights cross-domain generalization to diffusion-driven phenomena.

