# OpenReview forum: "Towards Universal Neural Operators through Multiphysics Pretraining"
_ICLR.cc/2026/Conference — ICLR 2026 Conference Withdrawn Submission_

### Official Review · Reviewer_xuEJ · 2025-10-27

**Soundness:** 3
**Presentation:** 3
**Contribution:** 2
**Rating:** 4
**Confidence:** 3

**Summary:**

The paper proposes a way to make neural operators more universal by pretraining across multiple physics, and then fine tune on a new PDE. The suggestion is to keep a shared stack of kernel integral operator blocks and, at the fine tune time, freeze this backbone, while training only small input/output adapters. This cuts adaptation cost and tests, whether the backbone captures reusable physics. The authors build a multiphysics data pipeline that standardizes heterogeneous PDE datasets, enabling joint pretraining over advection, Burgers, Navier–Stokes, etc. The training scheme is: pretrain the full model on mixed physics, then fine tune only adapters for the target task. Fig. 2 illustrates the setup. The authors find that: (i) Pretraining a backbone and fine tuning only the small adapters improves error and shortens epochs versus training the whole model from scratch (table 1), (ii) When the fine tune task augments the equation with an extra term, the pretrained backbone plus adapters still wins (table 2), (iii) Pretrain on one set of PDEs , e.g. advection and Burgers, then fine tune on another, e.g. reaction–diffusion, yields lower error and faster epochs (table 3 and figure 3).

**Strengths:**

(i) Clear recipe: freeze backbone and train adapters. The pretrain to finetune scheme is explicit: fix the common integral operator stack and update only small input/output adapters, which highlights what transfers and cuts training cost. (ii) Well-designed evaluation: the authors test three distinct scenarios, new parameters, added physics/inputs, and cross PDE transfer, without changing the modeling recipe, (iii) Consistent empirical gains:  across tables, pretraining plus adapters improves accuracy and reduces average epoch time versus scratch, sometimes by large margins.

**Weaknesses:**

(i) Scope limited to same dimensionality and curated grids: the method is explicitly evaluated when pretrain and fine tune tasks have the same problem dimensionality, and the pipeline resamples all data to a shared fixed grid. That leaves open transfer across 2D to 3D, irregular meshes, or complex geometries/boundaries, (ii) Speedups reported per epoch, not end to end:  tables report Avg. epoch time(s) but not the total wall clock including pretraining. This makes it hard to judge the true efficiency once pretraining cost is amortized, (iii) Missing controls/ablations on fine tuning strategy: fine-tuning always freezes the operator backbone and there is no baseline that unfreezes the backbone after pretraining or trains adapters with a randomly initialized frozen backbone to isolate how much of the gain comes from pretraining vs. just fewer trainable parameters, (iv) Evaluation metrics are generic and not physics aware: the core metric used is range normalized MAE (NMAE) and MSE, and  there is no explicit assessment of conservation laws, stability under long rollouts, or other physics diagnostics, so it is unclear how models behave far off the train distribution.

**Questions:**

Suggestions: (i) Report end to end efficiency and not just per epoch time. The tables list Avg. epoch time(s), but one cannot judge the total
wall clock once pretraining is amortized. Add: total pretrain hours, fine tune hours, and amortization break even curves, i.e  accuracy vs. total time/energy, and include FLOPs and trained parameter counts per setting, (ii) Ablate the fine tuning strategy by adding controls: (1) unfreeze the backbone (full fine tune), (2) train adapters atop a randomly initialized frozen backbone, (3) partial unfreezing, e.g. top k operator blocks.This isolates how much gain comes from pretraining vs. fewer trainable parameters, (iii) Go beyond shared regular grids and same dimensionality by adding tests on irregular meshes, varying domains/BCs, and 3D cases to demonstrate geometry robust transfer.

---

### Official Review · Reviewer_1i3M · 2025-10-30

**Soundness:** 3
**Presentation:** 2
**Contribution:** 2
**Rating:** 4
**Confidence:** 4

**Summary:**

The paper studied the possibility of pretrain on multi-physics data and finetune to downstream tasks, as the first stage of a foundation model for PDEs. The paper proposes two new architectures based on the Mamba and Perceiver, combined with FNO. The results show the new model has competitive performance.

**Strengths:**

- The paper studies foundation model for PDE, which is a very interesting and important problem.
- The paper systematically studied generalization across to new coefficients, forcing terms, and PDEs.
- Experiment shows pretraining is generally helpful.
- The new architectures seems to have better accuracy on extended equations and on new PDEs.

**Weaknesses:**

- Similar Pretraining stage has been studied in Poseidon and [1].
- One of the key question is not yet answered: is pretraining always helpful? Should we train on all the data (across PDEs, forcing terms, and coefficients)? or should we limit pretraining to a certain subset?
- The new architectures of combining Mamba and Perceiver are not very significant. It is unclear if these modifications are helpful.

[1] McCabe, Michael, et al. "Multiple physics pretraining for spatiotemporal surrogate models." Advances in Neural Information Processing Systems 37 (2024): 119301-119335.

**Questions:**

- In the paper, finetuning is only learning on the lifting layer and projection layer. It will be better to some ablation. Sometime it's better to only adapt the last several layers.
- In the experiment is ths Swin-v2 model exactly the same as the SCOT model in Poseidon paper? If so, it would be better to name it as SCOT or Poseidon.

---

### Official Review · Reviewer_N1Vh · 2025-11-01

**Soundness:** 2
**Presentation:** 3
**Contribution:** 2
**Rating:** 2
**Confidence:** 3

**Summary:**

This paper introduces a pre-training approach for transformer neural operators based on adapter fine-tuning, and showcase high generalization capabilities, including cross-equation transfers. Two architectures, Mamba FNO and the Perceiver, are studied in detail. Experiments emphasized the models’ high performance in comparison to other neural operators, in tasks such as inference for PDEs with out of sample parameter values, addition of terms in the input function and knowledge transfer to unseen physical problems.

**Strengths:**

-	Tackles the difficult problem of training foundation models applicable across a range of different PDEs and parameters.
-	Introduces two novel architectures allowing pre-training and efficient fine-tuning.
-	Studies a diverse benchmark of PDEs and groups of PDEs, showcasing the robustness of the models.

**Weaknesses:**

-	No ablation studies to showcase the importance of the adapters, or other layers such as the self-attention
-	Generally speaking, it does not seem that the gains in terms of accuracy and training time are significant, vis-à-vis other neural operator architectures allowing pre-training and fine-tuning, but even in comparison with FNO.

**Questions:**

-	In the tables, does the average per epoch include the pre-training for the fine-tuned cases ?
-	Could the authors provide some ablation studies to showcase the importance of the various architectural elements in their models ?

---

### Note · Authors · 2025-12-03

**Comment:**

We are grateful to the reviewers for the valuable comments. We understand that our paper is yet not ready and require significant changes and therefore we decided to withdraw it to submit to another venue.

**Withdrawal Confirmation:**

I have read and agree with the venue's withdrawal policy on behalf of myself and my co-authors.